# Effects of Oncolytic Vaccinia Viruses Harboring Different Marine Lectins on Hepatocellular Carcinoma Cells

**DOI:** 10.3390/ijms24043823

**Published:** 2023-02-14

**Authors:** Yanrong Zhou, Qianpeng Wang, Qi Ying, Xiaomei Zhang, Kan Chen, Ting Ye, Gongchu Li

**Affiliations:** College of Life Sciences and Medicine, Zhejiang Sci-Tech University, Hangzhou 310018, China

**Keywords:** oncolytic vaccinia virus, marine lectins, apoptosis, viral replication, signaling pathways

## Abstract

Oncolytic viruses are being developed as novel strategies for cancer therapy. Our previous studies have shown that vaccinia viruses armed with marine lectins improved the antitumor efficacy in diverse cancer types. The objective of this study was to assess the cytotoxic effects of oncoVV harboring *Tachypleus tridentatus* lectin (oncoVV-TTL), *Aphrocallistes vastus* lectin (oncoVV-AVL), white-spotted charr lectin (oncoVV-WCL), and *Asterina pectinifera* lectin (oncoVV-APL) on HCC. Our data revealed that the effects of recombinant viruses on Hep-3B cells were oncoVV-AVL > oncoVV-APL > oncoVV-TTL > oncoVV-WCL; oncoVV-AVL showed stronger cytotoxicity than oncoVV-APL, while oncoVV-TTL/WCL had no effect on cell killing in Huh7 cells, and PLC/PRF/5 cells exhibited sensitivity to oncoVV-AVL/TTL but not to oncoVV-APL/WCL. The cytotoxicity of oncoVV-lectins could be enhanced by apoptosis and replication in a cell-type-dependent manner. Further research revealed that AVL may mediate various pathways, including MAPK, Hippo, PI3K, lipid metabolism, and androgen pathways through AMPK crosstalk, to promote oncoVV replication in HCC in a cell-dependent manner. OncoVV-APL replication could be affected by AMPK/Hippo/lipid metabolism pathways in Hep-3B cells, AMPK/Hippo/PI3K/androgen pathways in Huh7 cells, and AMPK/Hippo pathways in PLC/PRF/5 cells. OncoVV-WCL replication was also multi-mechanistic, which could be affected by AMPK/JNK/lipid metabolism pathways in Hep-3B cells, AMPK/Hippo/androgen pathways in Huh7 cells, and AMPK/JNK/Hippo pathways in PLC/PRF/5 cells. In addition, AMPK and lipid metabolism pathways may play critical roles in oncoVV-TTL replication in Hep-3B cells, and oncoVV-TTL replication in Huh7 cells may depend on AMPK/PI3K/androgen pathways. This study provides evidence for the application of oncolytic vaccinia viruses in hepatocellular carcinoma.

## 1. Introduction

Liver cancer is currently the sixth-most common cancer and the third-most common lethal cancer in the world [1]. Of all primary liver cancer cases, hepatocellular carcinoma accounts for more than 90% [2,3]. Despite recent advances in hepatocellular carcinoma therapy, patients with recurring or late stages are still not considerably curable [4,5]. Therefore, treatment strategies are urgently required for patients suffering from hepatocellular carcinoma.

Oncolytic viruses are native or genetically engineered viruses, which selectively target cancerous cells while sparing healthy cells [6]. To date, a variety of oncolytic viruses have exhibited therapeutic potential in cancer, such as adenovirus, herpes simplex virus, measles virus, Newcastle disease virus, vaccinia virus, and reovirus [7,8]. Among these oncolytic viruses, vaccinia viruses have received more attention due to their strong transgene-encoding ability, efficient expression systems, stability, and safety record as a live vaccine in human history [9]. Moreover, expressing the optimal harbored genes may be a strategy to improve the antitumor effect of vaccinia virus.

Lectins are a class of specific glycoproteins that play essential roles in various biological processes, including cell recognition, innate immunity, and cell division, by preferentially recognizing and binding to carbohydrate complexes [10,11]. Previous studies reported that lectins derived from aquatic organisms presented the potential for cancer therapy. Intrinsic apoptosis pathways can be triggered by sialic acid-binding lectins obtained from *Rana catesbeiana* in many kinds of tumor cell lines [12,13]. *Eucheuma serra* agglutinin from marine algae exhibited the ability to induce cell apoptosis in a broad spectrum of cancer types, including colon adenocarcinoma, breast cancer, and osteosarcoma cells [14,15,16]. MytiLec lectin derived from *Mytilus galloprovincialis* triggered the loss of membrane integrity in human Burkitt lymphoma Raji cells [17,18].

The oncolytic vaccinia virus (oncoVV) used in this research is a Western Reserve (WR) strain engineered for viral thymidine kinase (TK) gene deletion [19]. Our previous studies have shown that oncoVV armed with *Tachypleus tridentatus* lectin (oncoVV-TTL) [20], *Aphrocallistes vastus* lectin (oncoVV-AVL) [21,22,23,24], white-spotted charr lectin (oncoVV-WCL) [25], and *Asterina pectinifera* lectin (oncoVV-APL) elicited cytotoxicity to a variety of cancer cells by enhancing viral replication through the suppression of the antiviral response of cells and inducing cell apoptosis. In this study, the cytotoxic effects of four vaccinia virus variants constructed from marine lectins were compared, and the mechanisms of action of these viruses were further investigated in HCC. These observations may contribute to the application of vaccinia viruses in cancer therapy.

## 2. Results and Discussion

### 2.1. The Cytotoxic Effect of OncoVV-Lectins in HCC

To investigate the cytotoxicity of oncoVV-TTL/AVL/APL/WCL, an MTT assay was performed in Hep-3B, Huh7, and PLC/PRF/5 cell lines with 5 or 10 MOI at 24, 48, and 72 h postinfection. In Figure 1, the marine lectins AVL/APL/TTL/WCL were effective at reducing the cell viability of HCC. Figure 1a shows that oncoVV and recombinant viruses exhibited cytotoxic effects on Hep-3B cells, with less than 100% cell viability. Compared to the PBS control, oncoVV-TTL/AVL/APL/WCL decreased cell viability to 24%, 5%, 11%, and 26% at 5 MOI for 72 h, suggesting that the rankings were oncoVV-AVL > oncoVV-APL > oncoVV-TTL > oncoVV-WCL according to their effectiveness in Hep-3B cells (Figure 1a). In Figure 1b, compared with the control PBS, both oncoVV-AVL (20%) and oncoVV-APL (62%) show strong cytotoxic effects on Huh7 cells at 72 h. However, the cell viability after oncoVV or oncoVV-TTL/WCL treatments was higher than that after PBS, suggesting that these viruses may act as proliferation stimulators in Huh7 cells, and the specific mechanism needed to be further explored (Figure 1b). As seen in Figure 1c, oncoVV and oncoVV-APL/WCL had no effect on cell killing for 72 h, but PLC/PRF/5 cells exhibited sensitivity to oncoVV-AVL (43%) and oncoVV-TTL (82%). In total, the effects of recombinant viruses on Hep-3B cells were oncoVV-AVL > oncoVV-APL > oncoVV-TTL > oncoVV-WCL. Recombinant viruses oncoVV-AVL showed stronger cytotoxicity than oncoVV-APL, whereas oncoVV-TTL/WCL had no effect on cell killing in Huh7 cells; PLC/PRF/5 cells exhibited sensitivity to oncoVV-AVL and oncoVV-TTL but not to oncoVV-APL/WCL.

### 2.2. Apoptotic Effect of OncoVV Armed with Marine Lectins in HCC

To verify the underlying mechanism of recombinant viruses’ cytotoxic effects, apoptosis was determined by flow cytometry. In Figure 2a, compared with the control, oncoVV-AVL and oncoVV-APL enhanced the apoptotic effect, with 35% (oncoVV-AVL) and 22% (oncoVV-APL). In Huh7 cells, the proportions of apoptotic cells exposed to oncoVV-TTL/AVL/APL were 5-, 15-, and 5-fold higher than those exposed to oncoVV, indicating that oncoVV-AVL had a more significant ability to induce apoptosis than oncoVV-APL and oncoVV-WCL (Figure 2b). Figure 2c shows that the apoptosis induced by oncoVV-TTL (13%) and oncoVV-AVL (28%) was higher than that induced by oncoVV, which indicates that TTL/AVL had the ability to induce Huh7 cells apoptosis. These results suggest that apoptosis plays a key role in the cytotoxicity of recombinant viruses against HCC.

### 2.3. Replication of OncoVV Harboring Marine Lectins Was Enhanced in HCC

To further study whether lectins enhanced viral replication, virus yields of oncoVV-TTL/AVL/APL/WCL were examined in HCC. A27L, as a protein located on the surface of the intracellular mature virus [26], was also evaluated. As shown in Figure 3, the virus yields of recombinant viruses were higher than that of oncoVV. In particular, oncoVV-AVL replicated significantly more quickly and strongly than the others. The replication of oncoVV-APL was significantly stronger than that of oncoVV-TTL/WCL in PLC/PRF/5 cells (Figure 3c), but the opposite result was observed in Huh7 cells (Figure 3a). Western blot analysis showed that the levels of A27L, which could be subject to a variety of modifications [26,27,28], were consistent with the results of viral replication (Figure 3b,d). The results demonstrated that arming oncolytic vaccinia viruses with marine lectins improved viral replication.

### 2.4. Pathways Involved in the Replication of OncoVV Armed with Marine Lectins

To investigate the effects of signal pathways on viral replication, a combination of vaccinia viruses with various inhibitors or activators was used. AMP-activated protein kinase (AMPK) is a heterotrimeric complex consisting of the catalytic subunit α and regulatory subunits β and γ [31]. In our study, AICAR (AMPK activator) was used to assess the viral replication in HCC. Our results revealed that AICAR markedly decreased the replication of both oncoVV and recombinant vaccinia viruses to varying degrees in Hep-3B and Huh7 cells (Figure 4a,b). In particular, AICAR almost completely suppressed oncoVV and oncoVV-WCL replication in Huh7 cells (Figure 4b). Additionally, AICAR significantly reduced the replication of oncoVV and oncoVV-AVL/APL/WCL, while it had no effect on oncoVV-TTL replication in PLC/PRF/5 cells, indicating that TTL caused vaccinia virus insensitivity to AICAR. These observations indicate that the activation of AMPK inhibited viral replication in HCC without cell-type-dependent specificity.

AMPK, as a cellular energy sensor, regulates various cellular processes. For example, AMPK negatively regulates receptor tyrosine kinase pathways to further target the downstream effectors ERK and JNK [32] and can regulate insulin receptor substrate-1, which is an activating factor for PI3K/Akt/mTOR signaling [33]; AMPK is an upstream regulator of the Hippo pathway, which promotes the phosphorylation of YAP [34]. In addition, the AMPK pathway plays a role in regulating various metabolites. Therefore, AMPK may be one of the switches that interacted with MAPK, Hippo, PI3K, and metabolic pathways. In our study, the activation of AMPK inhibited oncoVV and oncoVV-lectin replication in HCC, contrary to the finding that AMPK facilitated the vaccinia virus infection of healthy cells by modulating the actin cytoskeleton and micropinocytosis [35], suggesting that the signal regulation of AMPK varied between healthy cells and HCC, pending further investigation. However, oncoVV-TTL replication in PLC/PRF/5 cells was not affected by AICAR, indicating that TTL caused vaccinia virus insensitivity to AICAR. These observations indicate that the activation of AMPK inhibited vaccinia virus replication in HCC without cell-type-dependent specificity, which could be changed by some types of lectins.

Research increasingly shows that the mitogen-activated protein kinase (MAPK) pathway contributes to enhancing viral replication [36]. As members of the MAPK pathway, the Raf/ERK signaling pathway is required for viral replication [23,25,37]. Rodríguez et al. reported that ERK activators increased viral yields, whereas ERK silencing decreased viral multiplication [38]. Our previous study found that oncoVV-AVL interfered with the Raf/ERK pathway in Hela S3 cells [22]. In Figure 5a,d, although both Sorafenib (Raf inhibitor) and U0126 (MEK inhibitor) decreased oncoVV replication, the expression levels of both ERK and phosphorylated ERK (p-ERK) were not upregulated by oncoVV as compared with PBS, indicating that oncoVV replication in Hep-3B cells was independent of this pathway. Similarly, WB results showed that oncoVV-TTL/APL did not alter the phosphorylation of ERK (Figure 5d), suggesting that oncoVV-TTL/APL could not activate the Raf/ERK pathway in Hep-3B cells. In addition, despite oncoVV-WCL-activated ERK expression, Sorafenib and U0126 had no effects on oncoVV-WCL replication, suggesting that WCL did not rely on this pathway to promote replication. Conversely, Figure 5d shows that oncoVV-AVL stimulated ERK phosphorylation, so oncoVV-AVL may enhance replication by regulating the Raf/ERK pathway in Hep-3B cells.

As seen in Figure 5b,e, Sorafenib decreased oncoVV replication, but p-ERK expression was decreased under oncoVV treatment, suggesting that oncoVV replication in Huh7 cells was independent of the Raf/ERK pathway. Furthermore, Sorafenib or U0126 restrained the replication of oncoVV-TTL/APL/WL, while the WB results showed that ERK phosphorylation was not stimulated by oncoVV-TTL/APL/WL as compared to the control, suggesting that their replication was not dependent on this pathway in Huh7 cells. However, the replication of oncoVV-AVL was inhibited by both Sorafenib and U0126, and Figure 5e further demonstrated that oncoVV-AVL increased the expression of p-ERK, suggesting that oncoVV-AVL replication in Huh7 cells depended on the Raf/ERK pathway.

As shown in Figure 5c,f, oncoVV replication was restrained by Sorafenib or U0126, and oncoVV stimulated ERK phosphorylation, suggesting that oncoVV replication in PLC/PRF/5 was dependent on the Raf/MEK/ERK pathway to a certain extent. Figure 5f illustrates that oncoVV-TTL/APL/WCL did not affect the expression of phosphorylated ERK, suggesting that oncoVV-TTL/APL/WCL could not activate this pathway. On the contrary, oncoVV-AVL promoted ERK phosphorylation (Figure 5f), and Figure 5c shows that oncoVV-AVL replication was partly dependent on the Raf/ERK pathway, indicating that oncoVV-AVL replication was involved in this pathway.

Taken together, our current data illustrate that oncoVV-AVL positively regulated the Raf/ERK pathway to promote viral replication in HCC. However, oncoVV-TTL replication was independent of this pathway, inconsistent with previous research indicating that oncoVV-TTL replication was highly dependent on ERK activity in the liver cancer cell lines MHCC97-H and BEL-7404 [20], suggesting that the ERK pathway facilitated oncoVV-TTL production in a cell-type-dependent manner. Additionally, oncoVV-APL/WCL replication was not directly related to this pathway in HCC either. Therefore, the Raf/MEK/ERK pathway contributed greatly to oncoVV-AVL replication in HCC.

The c-Jun N-terminal kinase (JNK) signal transduction pathway is another important branch of the MAPK pathway, which plays an important role in various physiological processes. JNK translocates into the nucleus and phosphorylates downstream transcription factors, such as c-Jun, thereby modulating cellular transcription. Wang et al. discovered that the white spot syndrome virus hijacked the host JNK pathway via its immediate-early 1 protein to drive replication [39]. In our study, the JNK inhibitor SP600125 was used to assess the virus yields. The expression level of c-JUN, a key downstream effector of JNK, was also detected. In Figure 6a, SP600125 could affect the replication of the control and oncoVV-WCL but not that of oncoVV-TTL/AVL/APL; Figure 6d shows that the expression of c-JUN was induced by oncoVV and oncoVV-WCL, and oncoVV-WCL led to a higher level of c-JUN, indicating that WCL may activate the JNK pathway to prompt oncoVV replication in Hep-3B cells. Figure 6b shows that the JNK pathway was involved in viral replication; Figure 6d suggests that both oncoVV and recombinant viruses dramatically induced the expression of c-JUN, and oncoVV-AVL infection induced higher levels of c-JUN expression, indicating that AVL activated the JNK pathway to enhance viral replication in Huh7 cells. Figure 6c shows that the JNK pathway could affect oncoVV and oncoVV-AVL/APL/WCL replication in PLC/PRF/5 cells. Figure 6f shows that oncoVV-AVL/WCL dramatically induced the cellular levels of c-JUN compared to the others, indicating that replication in PLC/PRF/5 cells was dependent on this pathway. These results suggest that the JNK/c-JUN pathway played essential roles in oncoVV-WCL replication in Hep-3B cells, oncoVV-AVL replication in Huh7 cells, and oncoVV-AVL/WCL replication in PLC/PRF/5 cells.

In total, oncoVV replication in HCC was strongly associated with the activities of JNK or c-JUN. However, exogenous lectins could partly alter this dependency. For instance, oncoVV-TTL/AVL/APL replication in Hep-3B was independent of the JNK pathway, and similar results could be detected in PLC/PRF/5 cells infected with oncoVV-TTL. Moreover, although oncoVV-WCL markedly induced the expression of c-JUN compared with PBS, oncoVV-WCL replication could not be inhibited by SP600126 in Huh7 cells, suggesting that oncoVV-WCL replication was negatively regulated by the JNK pathway. Therefore, our data indicate that exogenous lectins could alter the oncoVV replication mechanism involved in the JNK pathway in a cell-type-dependent manner.

The Hippo pathway contributes to the virus infection of cells; both mammalian sterile 20-like kinase 1/2 (MST1/2) and YAP are critical kinases in the Hippo signaling pathway [23,40,41]. Figure 7a shows that XMU-MP-1 (MST1/2 inhibitor) decreased the replication of oncoVV-APL compared with that of oncoVV. It is well-known that the MST1/2-LATS cascade can negatively regulate the activation of YAP [42]. Consistently, lower YAP expression was detected in the oncoVV-APL treatment group (Figure 7d). Additionally, XMU-MP-1 significantly reduced oncoVV-AVL/APL/WCL replication in both Huh7 and PLC/PRF/5 cells (Figure 7b,c), and WB analysis showed that oncoVV-AVL/APL/WCL triggered YAP degradation (Figure 7e,f). In addition, the inhibitor XMU-MP-1 decreased the replication of oncoVV-TTL compared with that of oncoVV in Huh7 cells, but oncoVV-TTL did not alter the expression of YAP, indicating that the replication of oncoVV-TTL was not involved in Hippo pathway. Taken together, these results show that the Hippo-YAP pathway was associated with oncoVV-APL replication in Hep-3B and oncoVV-AVL/APL/WCL replication in Huh7 and PLC/PRF/5 cells.

Recent studies indicate that MST1/2 and YAP can inhibit innate immunity caused by the RIG-I/cGAS-TBK1/IKKε-IRF3 axis. For instance, YAP expression is positively correlated with Hepatitis B virus replication in HCC [43]. Molluscum contagiosum virus may enhance replication by increasing the expression of YAP/TAZ and suppressing TBK1, which is a major modulator of cytosolic nucleic acid detection and the regulation of antiviral defense [44,45]. However, our current data reveal that YAP expression was negatively correlated with the replication of oncoVV-APL in Hep-3B cells and oncoVV-AVL/APL/WCL in Huh7 and PLC/PRF/5 cells, indicating that AVL/APL/WCL enhanced the replication level of oncoVV but not by inhibiting the antiviral response. Marine lectins may inhibit YAP function to enhance replication in a cell-type-dependent manner, but the specific mechanism needed to be further explored.

Cellular phosphatidylinositol-3-kinase (PI3K), as a regulator of gene transcription, is involved in the assembly and budding of poxviruses [46]. To evaluate the effects of PI3K on the replication of oncoVV harboring different lectins, the PI3K inhibitor KY12420 was used. KY12420 restrained the replication of both oncoVV and oncoVV-AVL in Hep-3B cells (Figure 8a). As shown in Figure 8b, KY12420 promoted the replication of oncoVV while significantly reducing the replication of oncoVV-TTL/AVL/APL in Huh7 cells. KY12420 did not affect the replication of oncoVV-TTL but dramatically decreased the others’ replication in PLC/PRF/5 cells (Figure 8c). These observations suggest that TTL/AVL/APL may enhance replication by activating the PI3K pathway in Huh7 cells.

Extensive research has shown that the PI3K pathway benefits an invading virus that depends on cellular functions. Poxviruses usually utilize PI3K/AKT activity for assembly, growth, and morphogenesis [47]. In our study, the PI3K inhibitor restrained oncoVV replication in Hep-3B and PLC/PRF/5 cells, which is in agreement with other studies reporting that the inhibition of PI3K decreased late gene expression and slowed down vaccina virus maturation [46]. However, KY12420 promoted oncoVV replication in Huh7 cells. Meanwhile, KY12420 resisted the replication of oncoVV-TTL/AVL/APL, indicating that TTL/AVL/APL may enhance replication by activating the PI3K signaling pathway in Huh7 cells.

### 2.5. Capsaicin and EPI-001 Inhibited OncoVV-Lectin Replication in HCC

Viral infection often changes cellular metabolism to facilitate maximal viral replication. Fatty acid synthase is closely related to the replication and assembly of vaccinia viruses [48]. Capsaicin is an agonist targeting the transient receptor potentiovanin potential vanilloid 1 (TRPV1), which regulates lipid metabolism [49,50,51]. As seen in Figure 9a, capsaicin did not affect the replication of oncoVV and oncoVV-AVL but dramatically downregulated the replication of oncoVV-TTL/APL/WCL in Hep-3B cells. Additionally, all viral replication was dramatically repressed by capsaicin in Huh7 cells (Figure 9b). Meanwhile, capsaicin exerted inhibitory action against the replication of oncoVV and oncoVV-AVL/APL/WCL in PLC/PRF/5 cells (Figure 9c). It is possible that the replication of oncoVV-TTL/APL/WCL in Hep-3B cells was partly regulated by lipid metabolism.

EPI-001, which modulates peroxisome proliferator-activated receptor-gamma (PPARγ) as well as suppresses androgen receptor (AR) activity [52], showed a remarkable inhibitory effect on viral replication in Hep-3B cells (Figure 9d), indicating that lectins may enhance oncoVV replication independently of this pathway. In addition, EPI-001 significantly decreased the replication of recombinant viruses but not oncoVV in Huh7 cells (Figure 9e), suggesting that TTL/AVL/APL/WCL may enhance replication by modulating the AR-mediated pathway in Huh7 cells. Figure 9f shows that the replication of oncoVV and oncoVV-AVL/APL/WCL was inhibited by EPI-001, and oncoVV-TTL replication was not affected by the inhibitor, so lectins may enhance oncoVV replication in PLC/PRF/5 cells independently of this pathway.

Our data show that capsaicin did not affect oncoVV replication but dramatically decreased the replication of oncoVV-TTL/APL/WCL in Hep-3B cells, revealing that marine lectins may partly depend on lipid metabolism to enhance viral replication in Hep-3B cells. In addition, androgen plays a critical role in metabolic homeostasis, and the androgen receptor (AR) is relevant to various diseases and viral replication [53,54]. In addition, blocking the AR with an antagonist inhibited the replication of oncoVV-lectins but not oncoVV in Huh7 cells, suggesting that the expression of marine lectins in Huh7 may activate the AR-related cell signaling pathway to enhance oncoVV replication.

In summary, lectin AVL could mediate various signaling and metabolic pathways, including the MAPK, Hippo, PI3K/Akt, and androgen pathways through AMPK crosstalk, to promote the replication of oncoVV in HCC in a cell-dependent manner. Unlike the replication mechanism of oncoVV-AVL, other recombinant viruses could not be activated by the Raf/MEK/ERK signaling pathway to promote replication. Specifically, oncoVV-APL replication could be affected by AMPK/Hippo/lipid metabolism in Hep-3B cells, AMPK/Hippo/PI3K/androgen pathways in Huh7 cells, and AMPK/Hippo pathways in PLC/PRF/5 cells. The oncoVV-WCL replication was also multi-mechanistic, which could be affected by AMPK/JNK/lipid metabolism pathways in Hep-3B cells, AMPK/Hippo/androgen pathways in Huh7 cells, and AMPK/JNK/Hippo pathways in PLC/PRF/5 cells. In addition, AMPK and lipid metabolism pathways may play critical roles in oncoVV-TTL replication in Hep-3B cells, and oncoVV-TTL replication in Huh7 cells may depend on AMPK/PI3K/androgen pathways.

## 3. Materials and Methods

### 3.1. Cell Lines and Viruses

Human embryonic kidney cell line HEK 293A and human liver carcinoma cell lines Hep-3B, Huh7, and PLC/PRF/5 were purchased from the Chinese Academy of Sciences. Cells were cultured in DMEM (Gibco, Thermo Fisher Scientific, Waltham, MA, USA) supplemented with 10% or 15% FBS (Hyclone Laboratories, Dunedin, Otago, New Zealand) and 1% penicillin–streptomycin in an incubator at 37 °C. The WR strain of vaccinia virus was obtained from ATCC. The control oncoVV was constructed from a WR strain with a TK gene deletion via homologous recombination. The recombinant viruses oncoVV-AVL/APL/TTL/WCL were previously described [20,21,22,23,24,25]. Viruses were grown in HEK 293A cells, and virus stocks were prepared from cell lysates.

### 3.2. MTT Assay

The cell viability of Hep-3B, Huh7, or PLC/PRF/5 was assessed by MTT assay as previously described [30]. Cells were cultured for 12 h and then treated with PBS, oncoVV, or recombinant viruses at 5 or 10 MOI. MTT (5 mg/mL) was added to the wells at 24, 48, and 72 h postinfection. Then, 150 μL Dimethyl sulfoxide (DMSO) was added to the wells. The absorbance at 490 nm was measured using a microplate reader (Multiskan, Thermo Scientific, Waltham, MA, USA). Values were calculated as the percentage of the PBS control.

### 3.3. Flow Cytometry Analysis

Apoptosis was assessed by flow cytometry as previously described [30]. In brief, cell lines (Hep-3B, Huh7, and PLC/PRF/5) treated with PBS, oncoVV, and recombinant vaccinia viruses were collected and then stained with Annexin V-FITC and PI. The samples were analyzed by flow cytometry (AccuriC6, BD Biosciences, San Jose, CA, USA).

### 3.4. Viral Replication Assay

The viral titers were determined by TCID_50_ assay as previously described [29,30]. Cell lines (Hep-3B, Huh7, and PLC/PRF/5) were infected with viruses at 5 MOI for 0 h, 12 h, 24 h, 36 h, and 48 h. To study how lectins affect viral replication through signaling pathways, some activators or inhibitors (Selleck Chemicals LLC, Houston, TX, USA) were administered, and the drug was used 1 h prior to infection. Agents used in Hep-3B cells were AICAR (400 μmol/L), Sorafenib (3 μmol/L), U0126 (10 μmol/L), SP600125 (5 μmol/L), XMU-MP-1 (2 μmol/L), KY12420 (4.5 μmol/L), Capsaicin (100 μmol/L), and EPI-001 (100 μmol/L). The agents used in Huh7 cells were AICAR (400 μmol/L), Sorafenib (10 μmol/L), U0126 (10 μmol/L), SP600125 (5 μmol/L), XMU-MP-1 (2 μmol/L), KY12420 (4.5 μmol/L), Capsaicin (250 μmol/L), and EPI-001 (100 μmol/L). The agents used in PLC/PRF/5 cell lines were AICAR (400 μmol/L), Sorafenib (10 μmol/L), U0126 (10 μmol/L), SP600125 (5 μmol/L), XMU-MP-1 (2 μmol/L), KY12420 (4.5 μmol/L), Capsaicin (100 μmol/L), and EPI-001 (100 μmol/L).

### 3.5. Western Blot Analysis

Proteins were assessed by WB as previously described [30]. Cells (Hep-3B, Huh7, and PLC/PRF/5) infected with oncoVV or recombinant viruses for 36 h at 5 MOI were collected and lysed. Cytosolic protein samples were quantified using a BCA Protein Quantification Kit (Vazyme Biotech, Nanjing, Jiangsu, China). Proteins were separated on 15% SDS-PAGE (dodecyl sulfate, sodium salt-polyacrylamide gel electrophoresis) gels and transferred to PVDF membranes (Millipore, Bedford, MA, USA). Membranes were blocked with 5% skim milk and incubated with primary antibodies (1:1000) overnight at 4 °C, followed by incubation with the secondary antibodies (1:5000) for 1 h. Finally, proteins were detected on membranes.

### 3.6. Statistical Analysis

Differences between means were determined by Student’s *t*-test in two groups, or the one-way ANOVA test if more than two groups were tested. *p* < 0.05 or *p* < 0.01 was deemed statistically significant.

## 4. Conclusions

Lectin is exploited to exclusively bind to cancer cells and exerts antitumor activity through the induction of cell death and the inhibition of cell proliferation in different forms [55,56,57,58]. Previously, vaccinia virus was used as a delivery vector to express marine lectin genes, which had shown impressive anticancer activity. In this study, the cytotoxicity of four recombinant vaccinia viruses was compared, and their mechanisms were further explored. The results showed that lectins enhanced the antitumor effects by inducing apoptosis and promoting replication. The effects of recombinant viruses on Hep-3B cells were oncoVV-AVL > oncoVV-APL > oncoVV-TTL > oncoVV-WCL. In Huh7 cells, oncoVV-AVL showed stronger cytotoxicity than oncoVV-APL, while oncoVV-TTL/WCL had no effect on cell killing; PLC/PRF/5 cells exhibited sensitivity to oncoVV-AVL and oncoVV-TTL but not to oncoVV-APL/WCL. Marine lectins may mediate multiple pathways, including the MAPK, Hippo, PI3K/Akt, lipid metabolism, and androgen pathways through AMPK crosstalk, to promote oncoVV replication in HCC in a cell-dependent manner, suggesting that key molecules in relevant signaling pathways had the potential to synergize with oncoVV-lectins to improve therapeutic efficacy. By comparing the various pathways of four recombinant viruses, we inferred that the Raf/MEK/ERK pathway partly contributed to enhancing the cytotoxicity of oncoVV-AVL in HCC. This research lays the foundation for the application research of oncoVV-lectins, and more in-depth research on oncoVV-marine lectins needs to be conducted.

## Figures and Tables

**Figure 1 ijms-24-03823-f001:**
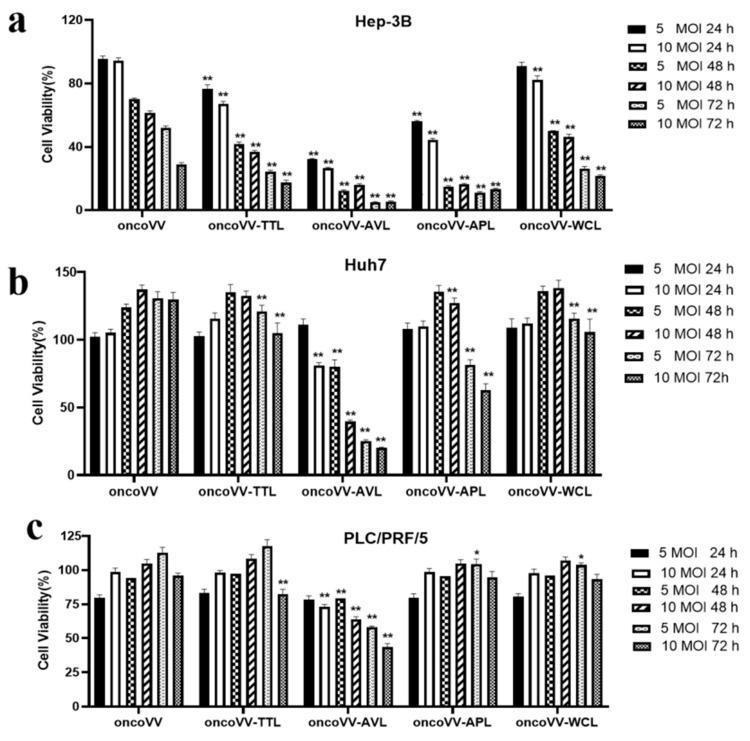
Percent viability (compared to PBS control) of HCC following infection with oncoVV-lectins. Cells were treated with PBS, oncoVV, or recombinant viruses at 5 or 10 MOI for 24, 48, or 72 h. Cell viability was measured by MTT assay in Hep-3B (**a**), Huh7 (**b**), and PLC/PRF/5 (**c**). OncoVV was used as the control. Values were calculated as the percentage of PBS control and presented as mean ± SEM (* *p* < 0.05, ** *p* < 0.01).

**Figure 2 ijms-24-03823-f002:**
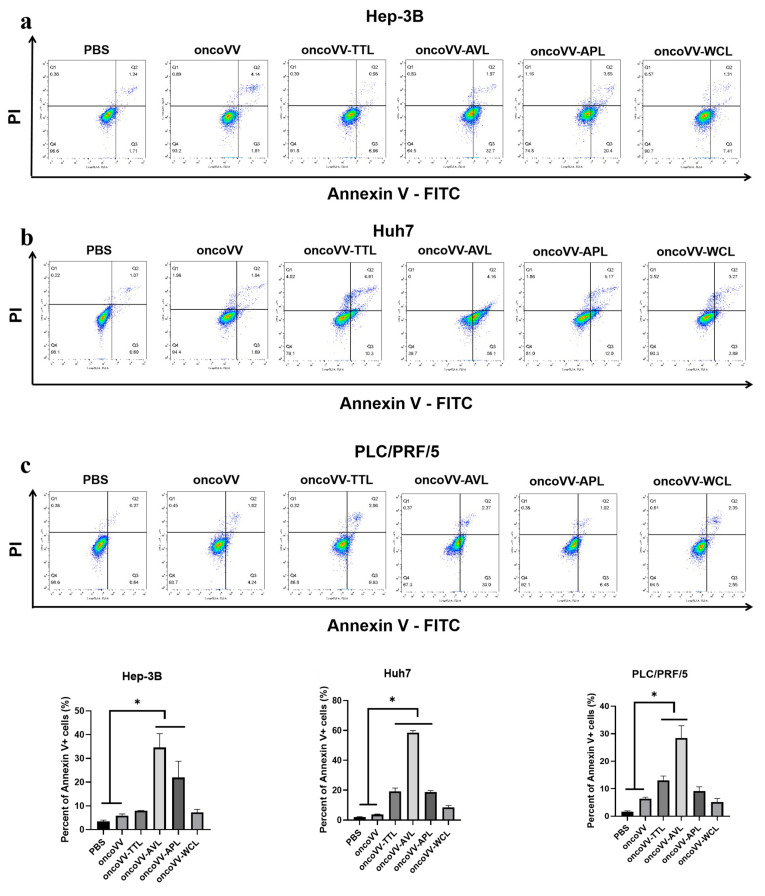
OncoVV-lectins induce apoptosis in Hep-3B (**a**), Huh7 (**b**), and PLC/PRF/5 (**c**) cells. Cells treated with PBS, oncoVV, or recombinant viruses (5 MOI) were harvested after 36 h, and then stained with annexin V-FITC and propidium iodide (PI). The apoptosis rates of Hep-3B (**a**), Huh7 (**b**), and PLC/PRF/5 (**c**) cells were immediately detected by flow cytometry. PBS and oncoVV served as controls (* *p* < 0.05).

**Figure 3 ijms-24-03823-f003:**
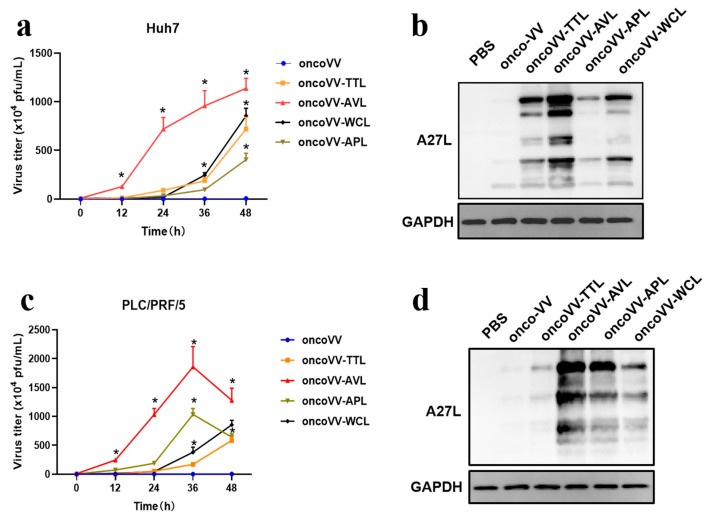
Lectins enhanced oncoVV replication in Huh7 (**a**,**b**) and PLC/PRF/5 (**c**,**d**) cells. Viral replication was determined by TCID_50_ assay. HEK293A cells (8 × 10^4^ cells/well) were cultured for 12 h and then infected with vaccinia viruses at 5 MOI. The viruses were collected at 0, 12, 24, 36, and 48 h, and then diluted successively (10^−1^~10^−9^). Diluted suspensions were then added to HEK 293A cells and viral plaque holes were counted. The formula T = 7 × 10^(d + 0.5)^ PFU/mL (d = sum of positive proportion) was used to calculate the viral titer [29,30]. The levels of A27L were detected by WB. PBS and oncoVV served as controls, and GAPDH was a loading control. (* *p* < 0.05).

**Figure 4 ijms-24-03823-f004:**
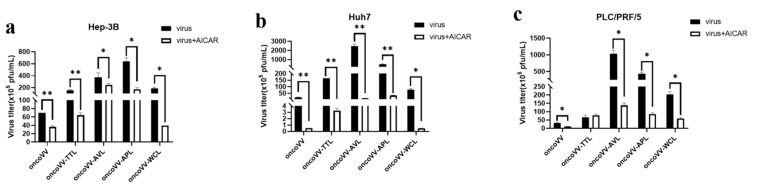
AlCAR affected vaccinia viruses replication in (**a**) Hep-3B, (**b**) Huh7, and (**c**) PLC/PRF/5 cells. Cells were maintained for 12 h and treated with AICAR (400 μmol/L), followed by 5 MOI of oncoVV or oncoVV-lectin infection about 1 h later. The above samples were collected at 36 h postinfection. OncoVV served as the control. * *p* < 0.05, ** *p* < 0.01.

**Figure 5 ijms-24-03823-f005:**
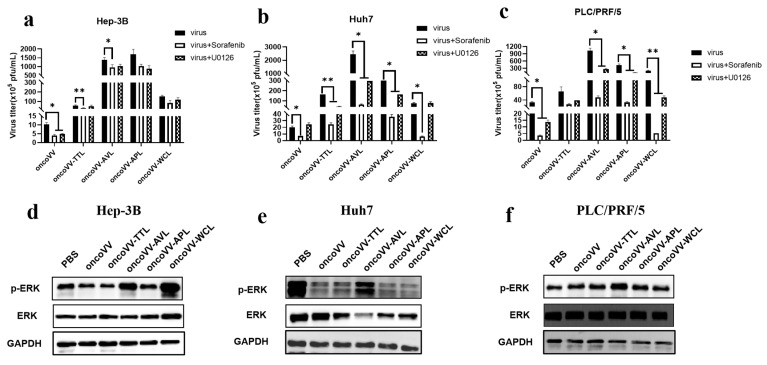
The effects of the ERK pathway on oncoVV-lectin replication in HCC. (**a**–**c**) Virus titers in the presence of Raf or MEK inhibitors (* *p* < 0.05, ** *p* < 0.01). Hep-3B, Huh7, and PLC/PRF/5 cells were maintained for 12 h and then treated with Sorafenib or U0126, followed by 5 MOI of recombinant virus infection about 1 h later. The above samples were collected at 36 h postinfection. Either 3 μmol/L Sorafenib or 10 μmol/L U0126 was used in Hep-3B cells; 10 μmol/L Sorafenib or 10 μmol/L U0126 was used in Huh7 and PLC/PRF/5 cells. (**d**–**f**) Expression levels of ERK and p-ERK determined by WB. Protein extracts after 36 h treatment were subjected to WB analysis for ERK and p-ERK. PBS and oncoVV served as the controls, and GAPDH was a loading control.

**Figure 6 ijms-24-03823-f006:**
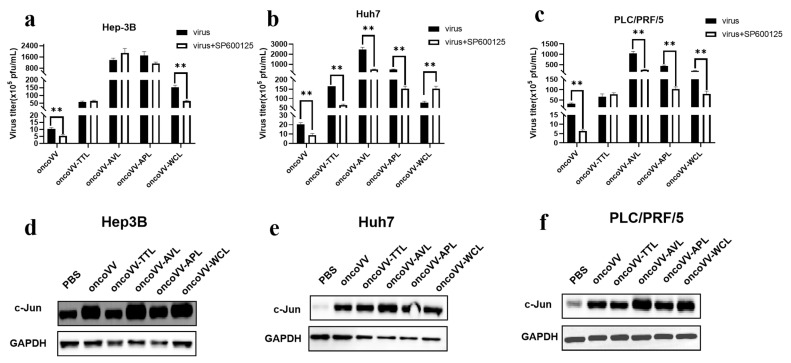
Relationships between JNK pathway and virus replication in HCC. (**a**–**c**) Virus titers in HCC treated with SP600125 (** *p* < 0.01). Hep-3B, Huh7, and PLC/PRF/5 cells were maintained for 12 h and treated with SP600125 (5 μmol/L), followed by 5 MOI of oncoVV or oncoVV-lectin infection about 1 h later. Samples were collected at 36 h postinfection. (**d**–**f**) The expression of c-JUN in Hep-3B (**d**), Huh7 (**e**), and PLC/PRF/5 (**f**) cells. c-JUN protein extracts after 36 h treatment were subjected to WB analysis. PBS and oncoVV served as the controls, and GAPDH was a loading control.

**Figure 7 ijms-24-03823-f007:**
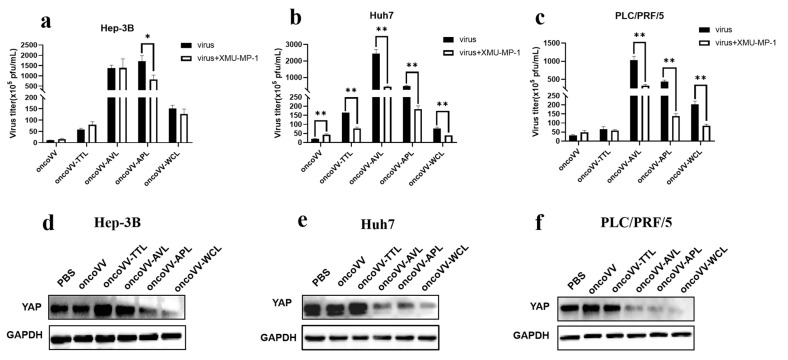
Relationships between Hippo/YAP pathway and virus replication in HCC. (**a**–**c**) Virus titers in the presence of XMU-MP-1 (* *p* < 0.05, ** *p* < 0.01). Hep-3B, Huh7, and PLC/PRF/5 cells were maintained for 12 h, treated with XMU-MP-1 (2 μmol/L), and infected with 5 MOI of recombinant viruses after 1 h. The samples were collected at 36 h postinfection. (**d**–**f**) The expression of YAP in Hep-3B (**d**), Huh7 (**e**), and PLC/PRF/5 (**f**) cells. The YAP protein extracts after 36 h treatment were subjected to WB analysis. PBS and oncoVV served as the controls, and GAPDH was a loading control.

**Figure 8 ijms-24-03823-f008:**
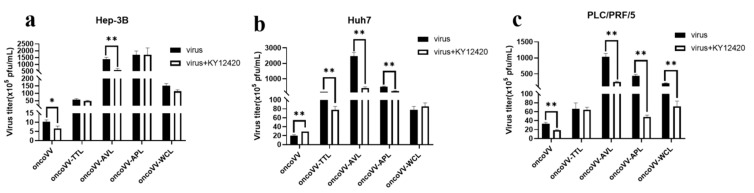
Virus yields in HCC treated with PI3K inhibitor. Hep-3B (**a**), Huh7 (**b**), and PLC/PRF/5 (**c**) cells were treated with combinations of oncoVV-lectins and KY12420 (4.5 μmol/L) at 5 MOI for 36 h. OncoVV served as a control (* *p* < 0.05, ** *p* < 0.01).

**Figure 9 ijms-24-03823-f009:**
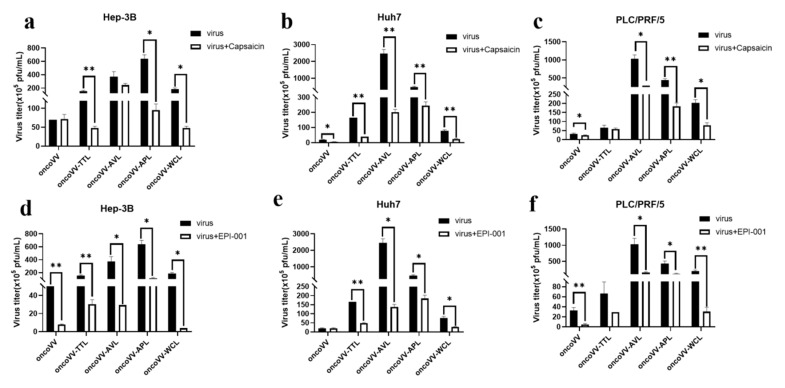
Virus yields in Hep-3B, Huh7, and PLC/PRF/5 cells. (**a**–**c**) Virus titers in HCC treated with capsaicin. (**d**–**f**) Virus titers in HCC treated with EPI-001. HCC were maintained for 12 h and treated with capsaicin or EPI-001, followed by 5 MOI of oncoVV or recombinant vaccinia virus. The samples were collected at 36 h postinfection (* *p* < 0.05, ** *p* < 0.01). OncoVV served as control. Either 100 μmol/L of capsaicin or 100 μmol/L of EPI-001 was used in Hep-3B and PLC/PRF/5 cells; 250 μmol/L of capsaicin or 100 μmol/L of EPI-001 was used in Huh7 cells.

## Data Availability

Not applicable.

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
