# Peer review of "Effects of Oncolytic Vaccinia Viruses Harboring Different Marine Lectins on Hepatocellular Carcinoma Cells"

_ijms, 2023, doi:10.3390/ijms24043823_

Round 1

Reviewer 1 Report

Gongchu Li and coworkers revealed that the importance of marine lectins TTL/AVL/APL/WCL in enhancement of the cytotoxicity of oncoVV in HCC by inducing apoptosis and promoting replication. This is a nice study, as is wont for this group very well executed and presented, building on previous work by the same group on development of strategies for oncolytic vaccinia viruses. The work stands out from a large number of marine lectins on hepatocellular carcinoma cells publications and shows that clever rational design can still lead to new discoveries. Authors are suggested to cite ‘’Org. Lett. 2018, 20, 9, 2611–2614’’ as one of the chemical methods for synthesis of Tumor-associated antigens in the introduction. In my opinion the work is very suited for publication in The International Journal of Molecular Sciences. It also invites for some follow‐up studies I would be interested to see in the future: oncoVV harboring marine lectins? To be sure, such studies are not required for accepting this work, which as mentioned I would accept as it is.

Author Response

Thanks for your kindly comments. The reference was added in the introduction section. (reference 59)

Reviewer 2 Report

There are many points need to improve the article

1-there are many abbreviations did not identify, please add abbreviation list (PBS, MOI, ----------)

2- in material and methods, where authors obtained the Onco viruses

3- Cytosolic proteins  samples were quantified (methods need)

4- In figure1b,c- Cell viability  of Huh7 (b), and PLC/PRF/5 was measured by MTT assay increased above 100%,  please add discussion

5-there are many references in results, we prefer combined results and discussion

6-Notice, do not use  (we in articles)

7- Also, sentence do not start with abbreviation

Reviewer 3 Report

The manuscript presented by Yanrong Zhou et al descried that marine lectins TTL/AVL/APL/WCL significantly enhanced the cytotoxicity of oncoVV in HCC by inducing apoptosis and promoting replication. Moreover, the authors found that marine lectins may mediate the multiple pathways, including MAPK,  Hippo, PI3K/Akt, lipid metabolism, or androgen through AMPK crosstalk, to promote oncoVV replication in HCC in a cell-dependent manner.

Although the topic was interesting, but some conclusions are not support by evidence. Below are some comments to help strengthen the manuscript.

1.    At line 64, “In this paper, we compared the antitumor effects of four vaccinia virus…”, however, the whole story did not show any anti-tumor experiment in vivo, such as, the tumor growth curve by oncoVVs.

2.    Legends are way too vague, do not emphasize the quality of their work don’t emphasize the quality of their work.

3.    For figure 1, lost 0 mol 24h and 48h as control. In figure 1b and c, why the cell viability (%) can over 100%?

4.    For Figure 2, The gate of flow data is strange. The compensation of Flow data need to do it. Such as, in figure 2a, the gate of the positive population of Annexin V in onvoVV-AVL was not right.

5.    In Materials and methods, there is no Flow cytometry analysis.

6.    English should be polished to improve quality of the work.

Round 2

Reviewer 3 Report

The authors addressed most of the questions.

Author Response

Dear reviewer,

Formatting and textual errors have been revised. Thank you for the advice.
